# MOD-ADAPTER: TUNING-FREE AND VERSATILE MULTI-CONCEPT PERSONALIZATION VIA MODULATION ADAPTER

**Weizhi Zhong**[1]***Huan Yang**[2], **Zheng Liu**[1], **Huiguo He**[5], **Zijian He**[1],
**Xuesong Niu**[2], **Di Zhang**[2], **Guanbin Li**[1,3,4]†

[1]School of Computer Science and Engineering, Sun Yat-sen University, Guangzhou, China
[2]Kolors Team, Kuaishou Technology
[3]Shenzhen Loop Area Institute, Shenzhen, China
[4]Guangdong Key Laboratory of Big Data Analysis and Processing, Guangzhou, China
[5]South China University of Technology, Guangzhou, China

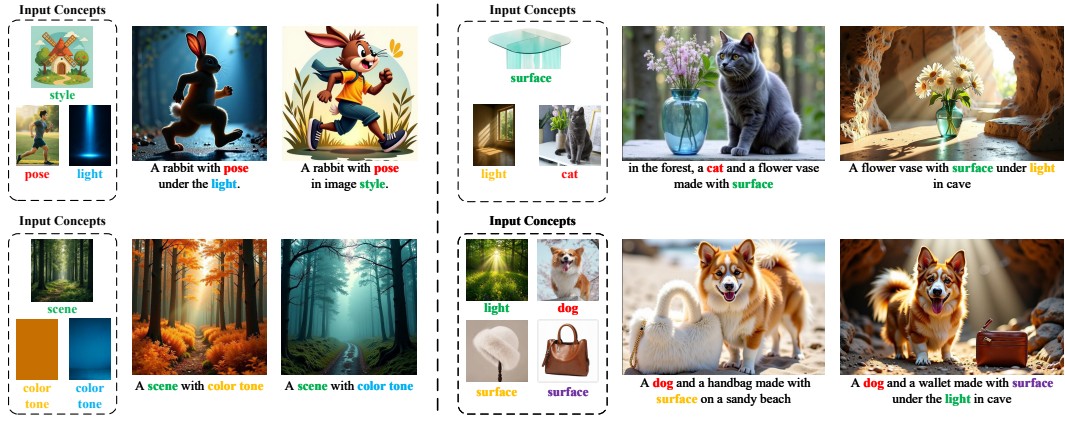

Figure 1: **Results of our multi-concept personalized image generation method.** Our method enables customizing both object and abstract concepts (e.g., pose, light, surface) without test-time fine-tuning. The colored words in the prompt below image indicate concepts to be personalized.

## ABSTRACT

Personalized text-to-image generation aims to synthesize images of user-provided concepts in diverse contexts. Despite recent progress in multi-concept personalization, most are limited to object concepts and struggle to customize abstract concepts (e.g., pose, lighting). Some methods have begun exploring multi-concept personalization supporting abstract concepts, but they require test-time fine-tuning for each new concept, which is time-consuming and prone to overfitting on limited training images. In this work, we propose a novel tuning-free method for multi-concept personalization that can effectively customize both object and abstract concepts without test-time fine-tuning. Our method builds upon the modulation mechanism in pre-trained Diffusion Transformers (DiTs) model, leveraging the localized and semantically meaningful properties of the modulation space. Specifically, we propose a novel module, Mod-Adapter, to predict concept-specific modulation direction for the modulation process of concept-related text tokens. It introduces vision-language cross-attention for extracting concept visual features, and Mixture-of-Experts (MoE) layers that adaptively map the concept features into the modulation space. Furthermore, to mitigate the training difficulty caused by the large gap between the concept image space and the modulation space, we introduce a VLM-guided pre-training strategy that leverages the strong image understanding

---

*Work done while interning at Kuaishou Technology.
†Corresponding author.

capabilities of vision-language models to provide semantic supervision signals. For a comprehensive comparison, we extend a standard benchmark by incorporating abstract concepts. Our method achieves state-of-the-art performance in multi-concept personalization, supported by quantitative, qualitative, and human evaluations. Project Page: `https://weizhi-zhong.github.io/Mod-Adapter`.

# 1 INTRODUCTION

Personalized text-to-image generation aims to synthesize images of the concepts specified by user-provided images in diverse contexts. Recently, this technique has attracted increasing research attention due to its broad applications, such as poster design and storytelling. However, existing personalized generation methods primarily focus on object concepts (e.g., common objects and animals) and struggle to personalize non-object/abstract concepts (e.g., pose and lighting), limiting their wider applicability. Recently, TokenVerse (Garibi et al., 2025) proposes a multi-concept personalization framework supporting both object and abstract concepts. However, its concept-specific test-time fine-tuning paradigm requires fine-tuning for each new concept image at test time, which is time-consuming and tends to overfit on the single training image, leading to suboptimal results. In this work, we take the first step toward a tuning-free framework enabling versatile multi-concept personalization for both objects and abstract concepts as shown in Fig.1.

Existing tuning-free personalized generation methods (Sun et al., 2024; Wang et al., 2025; Huang et al., 2025; Ma et al., 2024) often face two challenges when customizing abstract concepts. First, these methods failed to decouple the object concept and abstract concept from the input image due to the lack of an effective mechanism for extracting abstract features. As a result, they tend to directly replicate the object into the generated image. For example, when personalizing persons' pose concept, the generated person often closely resembles the one in the input concept image, rather than merely reflecting the pose features. This compromises the alignment between the generated image and the input prompt. Second, the features of abstract concepts are easily influenced by textual features or other concept features during generation, hindering the accurate preservation of the customized concept. This issue arises because these methods either concatenate concept image features with text features, or fuse them through additive cross-attention layers(Ye et al., 2023), resulting in limited localized control over the generated content.

To address these challenges, we propose a novel tuning-free framework for personalizing multiple concepts (both objects and abstract concepts) by leveraging the localized and semantically meaningful properties of the modulation space in DiTs. Specifically, we design a novel module, Mod-Adapter, to predict modulation directions for customized concepts. The directions are further integrated into the modulation process of concept-related textual tokens (e.g., "surface"), facilitating disentangled and localized control over the generated content. To effectively extract the target concept features from the input image, the Mod-Adapter introduces vision-language cross-attention layers that utilize the CLIP model's alignment capability between image and textual features. Besides, for accurately mapping the extracted concept visual features into the direction of DiT modulation space, we introduce a Mixture-of-Experts (MoE) mechanism within Mod-Adapter, where each expert is responsible for handling concepts with similar mapping patterns. Furthermore, to mitigate the difficulty of training Mod-Adapter from scratch due to the large gap between the concept image space and the DiT modulation space, we propose a VLM-guided pre-training strategy for better initialization, which leverages the strong image understanding capabilities of vision-language models to provide semantic supervision signals. To summarize, our key contributions are as follows:

- We propose a novel tuning-free and versatile multi-concept personalization generation method that can effectively customize both object and abstract concepts, such as pose, lighting, and surface, without requiring test-time fine-tuning for new concepts.

- We propose an innovative module, Mod-Adapter, to predict concept-specific personalized directions within the modulation space in a novel tuning-free paradigm. Within Mod-Adapter, the designed vision-language cross-attention extract concept visual features by leveraging the image-text alignment capability of CLIP, and the Mixture-of-Experts layers adaptively project these features into the modulation space. In addition, we propose a novel pre-training strategy guided by a vision-language model to facilitate training Mod-Adapter.

- We extend the commonly used benchmark by incorporating abstract concepts, resulting in a new benchmark named DreamBench-Abs. Experimental results demonstrate that our method achieves state-of-the-art performance in multi-concept personalization generation on this benchmark, as validated by quantitative and qualitative evaluations as well as user studies.

## 2 RELATED WORKS

In contrast to single-concept personalization works (Tan et al., 2025; Zhang et al., 2025) that can only customize a single concept, our discussion centers on multi-concept personalization, which can be further categorized into tuning-based and tuning-free methods.

**Tuning-Based Multi-concept Personalization.** Tuning-based multi-concept personalization approach (Choi et al., 2025; Jin et al., 2025; Kong et al., 2024; Kwon & Ye, 2025; Gu et al., 2023; Dong et al., 2024; Jiang et al., 2025; Kumari et al., 2023) requires one or more reference images of each concept to fine-tune the model before performing multi-concept personalization at test time. For example, Textual Inversion (Gal et al., 2023) and P+ (Voynov et al., 2023) introduce personalized text-to-image generation by learning a pseudo-word text embedding to represent each input concept. MuDI (Jang et al., 2024) and ConceptGuard (Guo & Jin, 2025) propose solutions to mitigate the concept confusion problem in multi-concept customization, using a form of data augmentation and concept-binding prompts techniques, respectively. Recently, TokenVerse (Garibi et al., 2025) proposed a disentangled multi-concept personalization method, which supports decoupling abstract concepts beyond objects, such as lighting conditions and material surfaces, from images. It optimizes a small MLP for each image to predict the modulation vector offsets for words in the image caption, learning to disentangle concepts of the image. Despite these advancements, tuning-based methods require test-time fine-tuning for unseen concepts, which is time-consuming and prone to overfitting on limited training images, often leading to suboptimal results. Contrasting with TokenVerse, we propose a novel tuning-free paradigm along with an innovative module, Mod-Adapter, and a novel pretraining strategy guided by a VLM.

**Tuning-Free Multi-concept Personalization.** Many studies (Sun et al., 2024; Ding et al., 2024; Wu et al., 2024; Huang et al., 2025; Shi et al., 2025; Chen et al., 2025) have attempted to explore tuning-free multi-concept personalized generation, which can generalize to unseen concepts without the need for test-time fine-tuning. Among these, Subject-Diffusion (Ma et al., 2024) and FastComposer (Xiao et al., 2024) integrate subject image features into the text embeddings of subject words. The combined features are then incorporated into the diffusion model through cross-attention layers to enable personalized generation for multiple subjects. Similarly, $\lambda$-ECLIPSE (Patel et al., 2024) projects image-text interleaved features into the latent space of the image generation model using a contrastive pre-training strategy. BLIP-Diffusion (Li et al., 2023a) follows a VLM BLIP-2 (Li et al., 2023b) to pre-train an encoder that produces text-aligned object-type concept representations. In contrast, our pre-training leverages the strong image understanding capabilities of a frozen VLM to facilitate the alignment of concept image features with the DiT modulation space. Benefit from the localized and semantically meaningful properties of the modulation space, our method can effectively customize both object and abstract concepts. Emu2 (Sun et al., 2024) proposes a generative autoregressive multimodal model for various multimodal tasks including multi-concept customization. InstructImagen (Hu et al., 2024a) introduces multimodal instructions and employs multimodal instruction tuning to adapt text-to-image models for customized image generation. MS-Diffusion (Wang et al., 2025) proposes a layout-guided multi-subject personalization framework equipped with a grounding resampler module and a multi-subject cross-attention mechanism. MIP-Adapter (Huang et al., 2025) introduces a weighted-merge mechanism to alleviate the concept confusion problem in multi-concept personalized generation. UniReal (Chen et al., 2025) proposes a unified framework for various image generation tasks, including multi-subject personalization, by learning real-world dynamics from large-scale video data. However, all existing tuning-free multi-concept personalization methods primarily focus on personalizing object concepts, and struggle to handle customization of abstract concepts. In this study, we propose a novel tuning-free framework for multi-concept personalization that can effectively customize both object and abstract concepts.

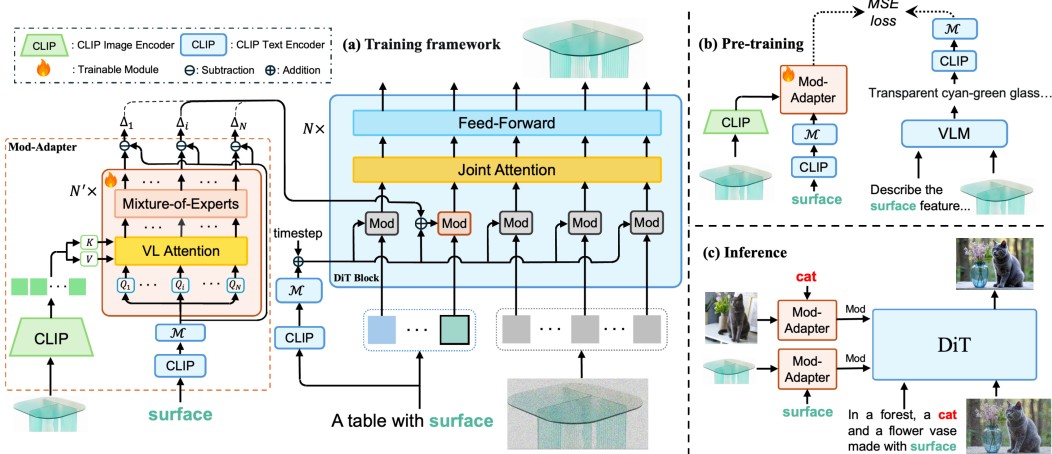

Figure 2: **Overview of the proposed method.** **(a)** During training, the proposed Mod-Adapter module takes as input a concept image and its corresponding concept word, and predicts a concept-specific modulation direction for each DiT block. The predicted directions are integrated into the modulation (Mod) process of the concept-related text tokens in DiT. **(b)** Pre-training of the Mod-Adapter module. The concept image is fed into a vision-language model (VLM) to obtain a detailed descriptive caption of the target concept in the image, which is further encoded by a CLIP text encoder and mapped by an MLP layer ($\mathcal{M}$) into the DiT modulation space. The resulting feature provides the semantic supervision signals for Mod-Adapter. **(c)** At inference, Mod-Adapter predicts a modulation direction for each customized concept. These directions are integrated into the modulation process of their corresponding text tokens to enable multi-concept customization.

## 3 METHOD

The overview of our proposed method is shown in Fig. 2. We propose a modulation space adapter module, named Mod-Adapter, for versatile multi-concept customization generation, along with a VLM-supervised adapter pretraining mechanism. In the following sections, we first introduce preliminaries on token modulation in DiTs (Sec. 3.1), and then detail the design of the Mod-Adapter module (Sec. 3.2) and the proposed pretraining strategy (Sec. 3.3).

### 3.1 PRELIMINARIES

Diffusion Transformers(DiTs) (Peebles & Xie, 2023) have recently emerged as a promising architecture for diffusion models (Ho et al., 2020), owing to the strong scalability of transformer (Vaswani et al., 2017). In text-to-image DiTs (Esser et al., 2024; Labs, 2024), text tokens and noisy image tokens are jointly processed through $N$ DiT blocks, each consisting of joint attention and feed-forward layers, to predict the noise added to the image VAE (Kingma & Welling, 2013) latent. In addition, the diffusion timestep condition and a global representation of the text prompt are integrated into the generation process via a token modulation mechanism before joint attention. In this work, we build on FLUX (Labs, 2024), a state-of-the-art DiT-based text-to-image model, for multi-concept personalization. Specifically, in each DiT block, all image and text tokens are modulated by a shared conditioning vector $y$ through Adaptive Layer Normalization (AdaLN) (Peebles & Xie, 2023). The modulation vector $y$ is computed by summing the diffusion timestep embedding $t_{emb}$ and a projection of the CLIP (Radford et al., 2021) pooled prompt embedding, as follows:

$$y = \mathcal{M}_t(t_{emb}) + \mathcal{M}(\text{CLIP}(p)), \tag{1}$$

where $p$ is the text prompt, $\mathcal{M}_t$ and $\mathcal{M}$ are two distinct MLP mapping layers. TokenVerse (Garibi et al., 2025) demonstrates that modulating individual tokens differently enables localized manipulations over the concept of interest during generation process. Specifically, instead of using the same modulation vector for all tokens, the text tokens associated with the target concept are modulated using adjusted modulation vectors:

$$y' = y + s\Delta_{attribute}, \tag{2}$$

where $s$ is a scale factor, the direction $\Delta_{attribute}$ captures the personalized attributes of the concept in the modulation space. The updated vectors $y'$ will induce localized effects on concept-related image regions through the joint attention layers. Thanks to the semantically additive properties of CLIP textual embedding, previous works (Baumann et al., 2025; Hu et al., 2024b; Garibi et al., 2025) have shown that the semantic direction of an attribute can be estimated using contrastive prompts with and without the specific attribute. Specifically, $\Delta_{attribute}$ can be approximated as:

$$\Delta_{attribute} \approx \mathcal{M}(\text{CLIP}(p^+)) - \mathcal{M}(\text{CLIP}(p^0)), \tag{3}$$

where $p^+$ is a positive prompt with some attribute added (*e.g.*, "transparent cyan-green glass surface"), $p^0$ is a neutral prompt without attribute (*e.g.*, "surface"), i.e., the concept word itself. Token-Verse (Garibi et al., 2025) proposes training a separate MLP for each image to predict personalized directions $\Delta_{attribute}$ for the concept in the image. However, it requires fine-tuning a model at test time for each new concept image. In contrast, we propose a tuning-free method for multi-concept customization that can generalize to unseen concepts without test-time fine-tuning, as detailed in the following sections.

## 3.2 MODULATION SPACE ADAPTER

As illustrated in Fig. 2(a), our proposed Mod-Adapter takes a concept image and its corresponding concept word as input, and predicts a personalized direction $\Delta_{attribute}$ in the modulation space. Since FLUX (Labs, 2024) contains $N$ DiT blocks, Mod-Adapter predicts a distinct modulation direction $\Delta_i$ for each block to enhance the model's expressiveness, forming the set $\{\Delta_i \mid i = 1, \ldots, N\}$. As illustrated in Fig. 2(a), in the $i$-th DiT block, only $\Delta_i$ will be added to the original modulation vector. Inspired by the formulation in Eq. 3, Mod-Adapter first predicts the attribute feature of the customized concept in the modulation space, denoted as $F_i^+$. Then, the personalized modulation directions $\{\Delta_i \mid i = 1, \ldots, N\}$ are computed as follows:

$$\Delta_i = F_i^+ - \mathcal{M}(\text{CLIP}(p^0)) \tag{4}$$

To obtain attribute feature $F_i^+$, the Mod-Adapter is designed with a vision-language cross-attention mechanism and a Mixture-of-Experts (MoE) component, as detailed below.

**Vision Language Cross-Attention.** The proposed Mod-Adapter fully exploits the cross-modal alignment capability of the CLIP model (Radford et al., 2021) between image and text features. Specifically, to extract the desired concept features from the input concept image, the corresponding concept word $p^0$ (*e.g.*, "surface") is first passed through the CLIP text encoder followed by the MLP mapping layer $\mathcal{M}$ to obtain a neutral feature(*i.e.* $\mathcal{M}(\text{CLIP}(p^0))$). To generate a personalized modulation direction for each of the $N$ DiT blocks, the neutral feature is further projected by a linear layer into $N$ queries, denoted as $Q_1, \ldots, Q_N$. Sinusoidal positional embeddings are added to these queries for distinguishing the direction of different DiT blocks. Meanwhile, we encode the input concept image using the CLIP image encoder and project the fine-grained features from the penultimate layer into key and value, denoted as $K$ and $V$, respectively. Then, cross-attention between the text and image features is computed using the following formula:

$$\text{Attention}(Q_i, K, V) = \text{Softmax}(\frac{Q_i K}{\sqrt{d}})V, \tag{5}$$

where $d$ is the dimension of key and $i = 1, \ldots, N$.

**Mixture of Experts.** After extracting the concept visual feature via vision-language cross-attention, the feature must be mapped into the modulation space of the pre-trained DiT model for effective integration. A straightforward approach is to use an MLP layer for this mapping. However, we find that this leads to suboptimal performance, possibly due to the fact that different types of concepts exhibit distinct mapping patterns. This suggests that concepts with similar mapping patterns should be handled by the same mapping function, while those with significantly different patterns should be processed separately. Motivated by this intuition, we introduce a Mixture-of-Experts (MoE) mechanism to adaptively map various concept visual features into the modulation space, where each expert corresponds to a distinct MLP mapping network. The core of the MoE lies in the routing network, which is responsible for assigning different inputs to different experts. A common practice is to use a learnable linear gating network to perform the routing. However, we find that this approach tends to suffer from the well-known imbalanced expert utilization problem, where many experts

remain underused during training, even when using a load balancing loss (Shazeer et al., 2017). To address this issue, we design a simple parameter-free routing mechanism based on the $k$-means clustering algorithm. Specifically, we perform $k$-means clustering over the neutral features of all concept words (i.e., $\mathcal{M}(\text{CLIP}(p^0))$) in the training dataset, with the number of clusters equal to the number of experts. Each resulting cluster corresponds to concepts of certain categories, which are assigned to a specific expert for processing.

**Training and Inference.** The training framework of Mod-Adapter is illustrated in Fig. 2(a). The proposed Mod-Adapter is the only component that requires training, while the pre-trained DiT-based text-to-image generation model is kept frozen. During training, only a single customized concept condition is added to the modulation space, while at inference time, multiple concept conditions can be added to the modulation of their corresponding concept tokens to enable multi-concept personalization, as illustrated in Fig. 2(c). We train Mod-Adapter using the same diffusion objective as the original DiT model (Labs, 2024). However, we find that training Mod-Adapter from scratch with this objective alone is challenging, possibly due to the large gap between the modulation space in DiT and the concept image space. To address this issue, we propose a VLM-supervised pretraining mechanism for Mod-Adapter to obtain a better initialization, as described below.

### 3.3 Mod-Adapter Pre-training Supervised by VLM

Our proposed pre-training approach is illustrated in Fig. 2(b), and is inspired by the formulations in Eq. 3 and Eq. 4 from the previous analysis. The attribute feature $F_i^+$ predicted by Mod-Adapter can be coarsely supervised during pre-training by the modulation-space representation (i.e., $\mathcal{M}(\text{CLIP}(p^+))$) of a positive prompt $p^+$ describing the attribute of the concept in image. To obtain an accurate positive prompt $p^+$ of the input concept image, we leverage a pretrained vision-language model (VLM) that already possesses strong image understanding capabilities. Specifically, the concept image and a pre-defined system prompt are fed into the VLM, where the system prompt guides the model to describe the detailed attributes of the target concept in the image, resulting in the output $p^+$. The positive prompt $p^+$ is then encoded by the CLIP text encoder and mapped by the MLP layer $\mathcal{M}$ into the modulation space. The resulting feature is used to supervise the Mod-Adapter's output $F_i^+$ during pre-training, with the MSE loss defined as:

$$\mathcal{L}_{pretrain} = \frac{1}{N} \sum_{i=1}^{N} \left\| F_i^+ - \mathcal{M}(\text{CLIP}(p^+)) \right\|_2^2 \tag{6}$$

During pre-training, only the objective $\mathcal{L}_{pretrain}$ is used, and the output of Mod-Adapter is not integrated into the memory-intensive DiT model, which enables efficient and lightweight pretraining. After pretraining, the Mod-Adapter is incorporated into the DiT model and trained further using only the diffusion objective mentioned above.

## 4 Experiments

### 4.1 Experimental Setups

**Datasets.** We train our model using the open-source MVImgNet object dataset (Yu et al., 2023), the Animal Faces-HQ (AFHQ) dataset (Choi et al., 2020), and synthetic data generated by the FLUX (Labs, 2024) model. MVImgNet contains multi-view images of real-world objects. We select objects of 40 commonly seen categories from it and use a single-view image per object for training. AFHQ is a high-quality animal face dataset containing common categories such as cats, dogs, and wildlife. For abstract concepts, we synthesize training data using the FLUX model, which has been demonstrated by recent works (Tan et al., 2024; Cai et al., 2025a) to be an effective diffusion self-distillation strategy. The resulting abstract concept dataset covers a range of common categories, including those used in TokenVerse (Garibi et al., 2025) (environment light, human pose, scene, and surface), as well as our additional extensions of image style and color tone. In total, our final training dataset contains 106,104 images paired with corresponding captions.

**Implementation Details.** We adopt the pre-trained FLUX.1-dev model as our DiT backbone containing $N = 57$ blocks. The Mod-Adapter contains $N' = 4$ blocks and the number of experts is set to 12 according to the ablation study. It is the only module that requires optimization during training, with a total parameter count of just 1,667.77M. For more details, please see Appendix C.

Table 1: **Quantitative comparison**. CP and PF score evaluates concept preservation and image-text alignment respectively. Their product CP·PF is a comprehensive evaluation score. CLIP-T evaluates the image-text alignment. All scores range from 0 to 1. ↑: higher is better.

| Methods | Multi-Concept | | | | Single-Concept | | | |
|---|---|---|---|---|---|---|---|---|
| | CP↑ | PF↑ | CP·PF↑ | CLIP-T↑ | CP↑ | PF↑ | CP·PF↑ | CLIP-T↑ |
| Emu2 (Sun et al., 2024) | 0.53 | 0.48 | 0.25 | 0.299 | **0.73** | 0.57 | 0.42 | 0.288 |
| MIP-Adapter (Huang et al., 2025) | 0.68 | 0.55 | 0.37 | 0.328 | 0.70 | 0.39 | 0.27 | 0.277 |
| MS-Diffusion (Wang et al., 2025) | 0.62 | 0.51 | 0.32 | 0.326 | 0.57 | 0.40 | 0.23 | 0.282 |
| TokenVerse (Garibi et al., 2025) | 0.56 | 0.56 | 0.31 | 0.316 | 0.58 | 0.66 | 0.38 | 0.312 |
| **Mod-Adapter(Ours)** | **0.70** | **0.89** | **0.62** | **0.330** | 0.61 | **0.89** | **0.54** | **0.315** |

**Comparison Methods.** We compare our method with state-of-the-art multi-subject personalization approaches, including tuning-free methods Emu2 (Sun et al., 2024), MIP-Adapter (Huang et al., 2025), and MS-Diffusion (Wang et al., 2025), as well as the tuning-based method TokenVerse (Garibi et al., 2025). Since the training datasets used by MIP-Adapter and MS-Diffusion do not include abstract concept data, we fine-tune their released model weights using our abstract concept data to ensure a fair comparison. Emu2 has not released its training code, but its training data includes common abstract concepts such as style and lighting.

**Evaluation Benchmarks.** Following prior work (Wang et al., 2025; Huang et al., 2025; Sun et al., 2024), we evaluate the performance of both single-concept and multi-concept personalization on the DreamBench benchmark (Ruiz et al., 2023), which contains 30 object or animal concepts and 25 template prompts. In addition, we extend the DreamBench by incorporating 20 abstract concepts for a more comprehensive evaluation, resulting in a new benchmark named DreamBench-Abs. For multi-concept evaluation, we follow TokenVerse (Garibi et al., 2025) to construct 30 combinations from all 50 concepts. None of the test images or prompts appear in our training set. Among all methods, only TokenVerse, based on per-sample training, requires using the test images for training.

**Metrics.** We follow prior methods (Wang et al., 2025; Huang et al., 2025; Sun et al., 2024; Garibi et al., 2025) to evaluate both single-concept and multi-concept personalization from two perspectives: the fidelity of the generated concepts (Concept Preservation) and the alignment between the generated images and the textual prompt (Prompt Fidelity). As our task involves both object and abstract concepts, we follow TokenVerse (Garibi et al., 2025) and adopt a multimodal LLM-based (OpenAI, 2024; Peng et al., 2025) scoring approach. For each generated image, the multimodal LLM outputs a Concept Preservation (**CP**) and a Prompt Fidelity (**PF**) score, which evaluate concept preservation and image-text alignment, respectively. Since there is often a trade-off between the two metrics, their product (**CP·PF**) is also reported as a comprehensive score (Peng et al., 2025). Besides, we evaluate image-text alignment by measuring the similarity between their CLIP embeddings (**CLIP-T**).

## 4.2 COMPARISONS

**Quantitative Comparison.** We report the quantitative comparison results in Tab. 1. In multi-concept personalization, our method outperforms all previous approaches across all metrics. Specifically, it achieves the highest CP·PF score of 0.62, demonstrating a substantial +67.6% improvement over the second-best method, MIP-Adapter (0.37). While MIP-Adapter and MS-Diffusion achieve competitive CP scores (0.68 and 0.62, respectively), their PF scores (0.55 and 0.51) are significantly lower than ours (0.89). Consistent with observations from prior work, the CLIP-T metric is insensitive to variations in image-text alignment, with all methods scoring around 0.3. In single-concept personalization, our method still achieves significantly higher performance on the combined metric CP·PF compared to all other methods. Although Emu2 and MIP-Adapter outperform us on the CP metric, they perform significantly worse on both PF and CLIP-T, indicating that they sacrifice prompt fidelity to achieve higher concept preservation. In addition, while Emu2 performs well in single-concept personalization, its performance drops notably in the multi-concept setting.

**Qualitative Comparison.** Fig. 3 presents representative qualitative comparisons between our method and other approaches. In the first row of single-concept personalization, our method successfully generates a wallet with a brown leather surface consistent with the input concept image. In contrast, MS-Diffusion, MIP-Adapter, and Emu2 fail to effectively disentangle the abstract concept 'brown leather' from the object (handbag). As a result, they simply replicate the original object in the generated image, producing an undesired brown leather handbag instead of the desired wallet. This observation aligns with their high CP scores but low PF scores in Tab. 1. In the multi-subject per-

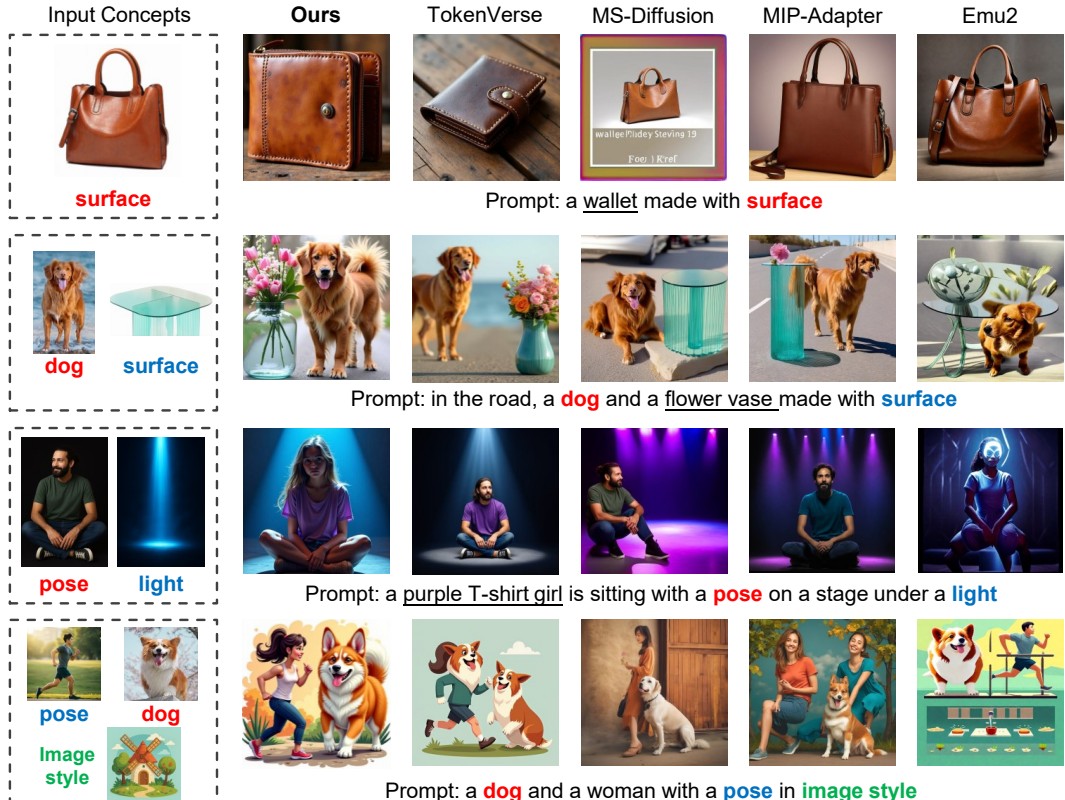

Figure 3: **Qualitative comparison.** The left dashed box shows input concept images. Colored words in the prompt indicate concepts to be personalized, while underlined text highlights elements that reflect differences in prompt alignment performance between methods.

Table 2: **User study results.** CP and PF respectively record the average scores given by volunteers for concept preservation and image-text alignment. Scores range from 1 to 5. ↑: higher is better.

| Methods | Multi-Concept | | Single-Concept | |
|---|---|---|---|---|
| | CP↑ | PF↑ | CP↑ | PF↑ |
| Emu2 (Sun et al., 2024) | 2.10 | 2.02 | 2.66 | 3.04 |
| MIP-Adapter (Huang et al., 2025) | 2.83 | 2.78 | 2.53 | 2.14 |
| MS-Diffusion (Wang et al., 2025) | 3.16 | 3.14 | 2.42 | 2.60 |
| TokenVerse (Garibi et al., 2025) | 3.35 | 3.48 | 3.43 | 2.87 |
| **Mod-Adapter(Ours)** | **4.29** | **4.40** | **4.49** | **4.60** |

sonalization setting, our method continues to demonstrate superior concept preservation and prompt alignment performance, whereas Emu2 shows reduced effectiveness, often generating unnatural concept combinations. MS-Diffusion and MIP-Adapter are still prone to "copy-paste" artifacts (e.g., the glass table in the second row and the man in the third row), which negatively affect prompt alignment. Meanwhile, their concept preservation performance also declines. In both single- and multi-concept personalization settings, tuning-based methods TokenVerse tend to overfit due to the need for fine-tuning on each input concept image, which compromises both concept preservation and prompt alignment. Its per-sample training paradigm only encourages the model to better reconstruct the training images with the input of corresponding image captions, but it does not ensure good performance on unseen test prompts in real-world scenarios. All these results demonstrate the superior performance of our method in multi-concept personalized generation.

**User Study.** Following the evaluation setting of TokenVerse (Garibi et al., 2025), we conducted a user study with 32 participants, asking them to rate each generated image in terms of concept preservation (CP) and prompt fidelity (PF). Each participant evaluated five results per method on both single- and multi-concept personalization settings, resulting in 4000 votes. The results are reported in Tab. 2. Our method receives consistently higher user ratings than all compared methods in both CP and PF. For more details about the user study, please see Appendix D.

## 4.3 ABLATION STUDY

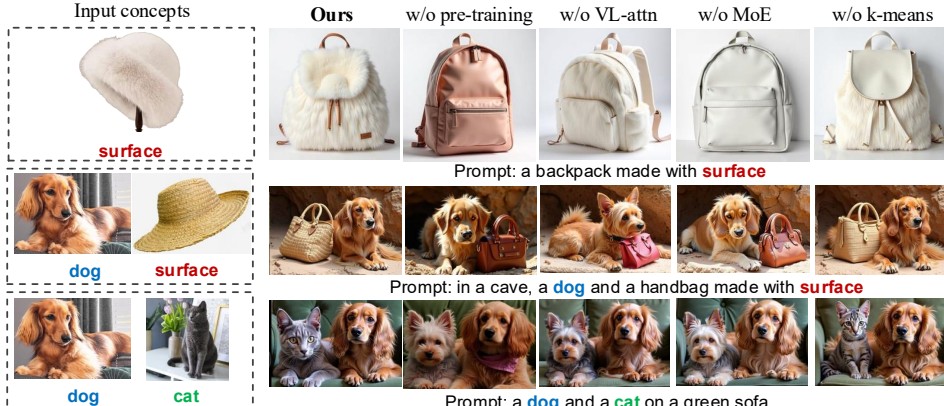

Figure 4: **Qualitative ablation results.** The left dashed box shows input concept images. Eliminating any proposed component degrades qualitative performance.

Table 3: **Quantitative ablation results.**

| Methods | Multi-Concept | | | | Single-Concept | | | |
|---|---|---|---|---|---|---|---|---|
| | CP↑ | PF↑ | CP·PF↑ | CLIP-T↑ | CP↑ | PF↑ | CP·PF↑ | CLIP-T↑ |
| w/o k-means routing | 0.59 | 0.83 | 0.49 | 0.324 | 0.54 | 0.82 | 0.44 | 0.311 |
| w/o MoE | 0.52 | 0.68 | 0.35 | 0.313 | 0.51 | 0.82 | 0.42 | 0.311 |
| w/o VL-attn | 0.52 | 0.75 | 0.39 | 0.329 | 0.57 | 0.86 | 0.49 | **0.317** |
| w/o pre-training | 0.29 | 0.58 | 0.17 | 0.306 | 0.33 | 0.72 | 0.24 | 0.308 |
| **Mod-Adapter(Ours)** | **0.70** | **0.89** | **0.62** | **0.330** | **0.61** | **0.89** | **0.54** | 0.315 |

**Mod-Adapter Pre-training.** We design an ablation variant excluding the Mod-Adapter pre-training strategy. Specifically, we train the Mod-Adapter from scratch using only the diffusion objective, under the same training settings and for the same total number of steps (176K) as the full model. The quantitative results of this variant are shown in the "w/o Pre-training" row of Tab. 3. Both CP and PF scores drop significantly for both single-concept and multi-concept personalization. Furthermore, as illustrated in the "w/o pre-training" column of Fig. 4, the generation quality degrades in terms of both concept preservation and prompt fidelity compared to our full model. This degradation is due to the large gap between the modulation space in DiT and the concept image space. In contrast, our proposed VLM-supervised pre-training mechanism effectively mitigates this training difficulty by leveraging the strong understanding capabilities of the VLM to provide an initialization.

**Vision Language Cross-Attention.** We design an ablation variant removing VL cross-attention from the Mod-Adapter module. In this variant, the concept word is not used as input, and the VL-attention is replaced with a standard cross-attention layer, where the queries are $N$ learnable query tokens following MS-Diffusion (Wang et al., 2025). As shown in the "w/o VL-atten" results in Tab. 3 and Fig. 4, this variant shows slightly degraded performance in both concept preservation and prompt fidelity in single-concept personalization. However, in the multi-concept setting, the performance drop is more significant, possibly due to the learnable query mechanism's inability to effectively extract target concept features. As mentioned earlier, the CLIP-T metric is insensitive to variations in prompt alignment; thus, the CLIP-T score of this variant remains comparable to, or even slightly higher than that of our full model.

**Mixture of Experts.** We design an ablation variant by replacing the MoE layer with a single MLP, while keeping the overall parameter count of the Mod-Adapter unchanged. As shown in the "w/o MoE" results on Tab. 3 and Fig. 4, this variant performs worse than our full model in both concept preservation and prompt fidelity. This is because a simple MLP is insufficient to accurately project diverse concept features into the modulation space of the pre-trained DiT network. For more ablation analysis on the number of experts, please refer to the Appendix F.

**K-means MoE Routing.** We design a variant following the common practice of using a learnable linear gating network for routing. As shown in Tab. 3 and Fig. 4, compared to our full model, this variant shows degraded performance in both concept preservation and prompt fidelity, but it still performs better than the "w/o MoE" variant. This is because the learnable linear gating network tends

to result in the under-utilization of certain experts, which is equivalent to using fewer experts than our full model. For more detailed analysis and the ablation on the clustering method, please refer to the Appendix F.

## 5 CONCLUSION

We propose a novel tuning-free framework for versatile multi-concept personalization, capable of customizing both object and abstract concepts without test-time fine-tuning. Our method contains a novel module, Mod-Adapter, consisting of vision-language cross-attention layers for concept visual feature extraction and Mixture-of-Experts layers for projecting features into the modulation space. Additionally, we introduce an innovative VLM-guided pretraining strategy to facilitate Mod-Adapter training. We conduct extensive experiments and demonstrate the superiority and effectiveness of our proposed method. For the limitation discussion of our method, please refer to the Appendix A.

## 6 ACKNOWLEDGMENTS

This paper is sponsored by CCF-Kuaishou Large Model Explorer Fund (No. CF-KuaiShou 2024007), and is also supported by the National Natural Science Foundation of China (No. 62322608), and supported by Kuaishou Technology.

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

# APPENDIX

## A  LIMITATIONS DISCUSSION

First, similar to TokenVerse (Garibi et al., 2025), our method may fail when two customized concepts have the same name in the prompt—for example, generating an image based on a prompt where the word "dog" appears twice. Second, when personalizing more than three concepts simultaneously, our method tends to deviate from the concept conditions. At the time of submission, our model was trained only on single-concept data, which may be the primary cause for the performance drop in multi-concept customization with an extreme number of concepts. Additionally, our framework relied solely on the CLIP image encoder for image feature extraction, which captures high-level features but tends to lose low-level object details. We present a potential solution: we synthesized training data involving two concepts (abstract and object) using the FLUX model, and combined it with the open-source MuSAR-Gen (Guo et al., 2025) multi-concept dataset, resulting in a total of 52,000 multi-concept samples for training. Furthermore, to mitigate the loss of low-level object details, we concatenated the VAE latents of concept images with the noise latents as additional conditions. We also introduced UnoPE (Wu et al., 2025) positional encoding to distinguish VAE latents from multiple reference images and applied lightweight LoRA fine-tuning to the joint attention layers in DiT. We present qualitative results for customizing 4 to 5 concepts in Figure 10. We leave addressing these limitations as future work.

## B  MORE RELATED WORKS

**Adapter.** Adapters offer an efficient paradigm for customized generation. Several adapter-based methods (Zhang et al., 2023; Mou et al., 2024; Ye et al., 2023; Zhao et al., 2023; Li et al., 2024; 2025) employ lightweight modules for task-specific adaptation while keeping the foundation model weights frozen. For example, ControlNet (Zhang et al., 2023) proposed a trainable copy module of its U-Net-encoder to incorporate spatial control conditions (e.g., edge maps). IP-adapter (Ye et al., 2023) proposed a decoupled cross-attention module to enable the image prompt control. Based on it, MIP-Adapter (Huang et al., 2025) further introduces a weighted-merge mechanism to alleviate the subject confusion problem in multi-subject personalized generation. In this paper, we propose a modulation space adapter module, Mod-Adapter, to predict concept-specific modulation directions while keeping the parameters of the pre-trained DiT backbone frozen.

## C  MORE IMPLEMENTATION DETAILS

We use Qwen2.5-VL-7B-Instruct (Team, 2025) as the vision-language model (VLM) for attribute caption and image caption. We train the model using the AdamW (Kingma & Ba, 2014) optimizer with a learning rate of $1 \times 10^{-4}$ on 8x 80GB GPUs. Mod-Adapter is first pre-trained for 50K steps with a batch size of 32 without being integrated into the DiT model, followed by an additional 126K steps of further training with a batch size of 1. Our method has the advantage of supporting multi-concept personalization at inference time while training only on single-concept data, especially given the difficulty of obtaining high-quality multi-concept training data. This ability stems from the localized and composable nature of the DiT modulation space. The scale factor $s$ in Eq. 2 is set to 1 during training and testing.

## D  USER STUDY DETAILS

Fig. 5 shows the user study interface for evaluating single- (a) and multi-concept (b) personalization. To assess concept preservation, participants were asked to rate the similarity between each generated concept and the corresponding concept in the reference image. For multi-concept cases, each concept was evaluated separately, and the final score was calculated by averaging the scores across all concepts. Participants selected from five options, each corresponding to a score from 1 to 5: "Very inconsistent" (1), "Somewhat inconsistent" (2), "Fair" (3), "Quite consistent" (4), and "Very consistent" (5). To assess prompt fidelity, participants evaluated how well the content of the generated image matched the given text prompt, using the same five-point scale.

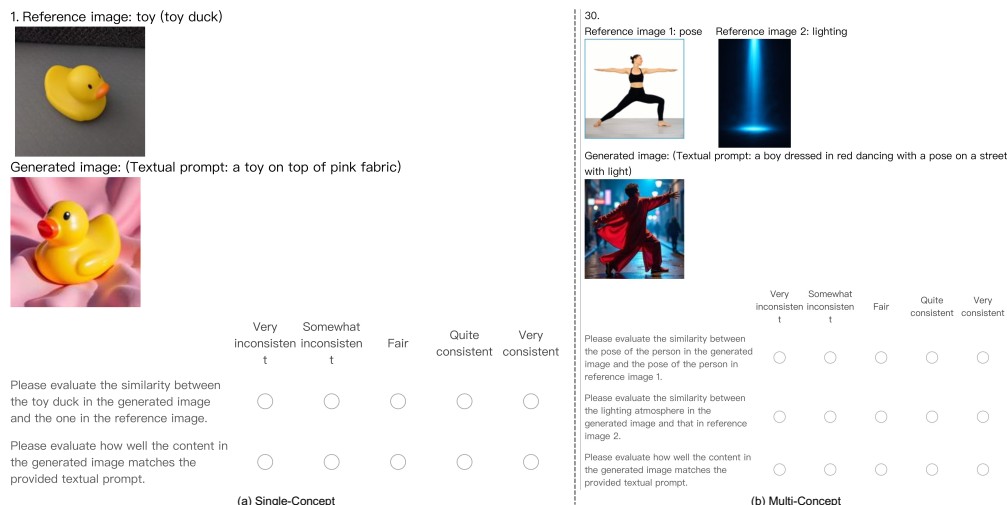

Figure 5: **Screenshot of our user study rating interface.** (a) Single-concept personalization evaluation. (b) Multi-concept personalization evaluation.

Table 4: More quantitative ablation results.

| Methods | Multi-concept | | | | single-concept | | | |
|---|---|---|---|---|---|---|---|---|
| | CP | PF | CP·PF | CLIP-T | CP | PF | CP·PF | CLIP-T |
| Ours (N_expert=1) | 0.40 | 0.54 | 0.22 | 0.272 | 0.36 | 0.48 | 0.17 | 0.278 |
| Ours (N_expert=4) | 0.43 | 0.59 | 0.25 | 0.305 | 0.40 | 0.78 | 0.31 | 0.310 |
| Ours (N_expert=8) | 0.57 | 0.65 | 0.37 | 0.310 | 0.50 | 0.79 | 0.40 | 0.308 |
| Ours (N_expert=12) | 0.70 | **0.89** | 0.62 | 0.330 | 0.61 | **0.89** | **0.54** | **0.315** |
| Hierarchical Clustering (N_expert=12) | **0.72** | 0.88 | **0.63** | **0.331** | **0.62** | 0.87 | 0.54 | 0.313 |

## E  BROADER IMPACTS

The development of the proposed versatile multi-concept personalization generation technique holds potential for broad societal benefits, such as facilitating personalized image creation, storytelling, poster design, and other creative applications. However, this technology may also raise concerns about potential negative societal impacts, such as its misuse to generate misleading or deceptive content, or enabling style-specific personalization that may infringe upon artists' rights. These risks can be mitigated through techniques such as watermarking generated images and developing robust deepfake detection algorithms (e.g. (Cai et al., 2025b; Hawkins et al., 2025)) for content authentication.

## F  MORE ABLATION ANALYSIS

**The number of experts and the clustering method.** In our MoE routing design, we suspect that concepts within the same cluster may share similar mapping patterns and should therefore be handled by the same expert. Based on this assumption, we set the number of clusters equal to the number of experts, such that each expert is responsible for processing concepts of multiple categories within one cluster. This design choice is empirically validated by our ablation study. Besides, we conducted experiments to investigate the impact of the number of clusters/experts (N_experts) and the choice of clustering method (k-means vs. hierarchical clustering). The results are shown in Tab.4. As observed, increasing the number of experts (i.e., the number of clusters) leads to improved model performance. We used 12 experts in our setup, the maximum that could be accommodated simultaneously within the 80GB GPU memory constraint. To accommodate more experts, the CPU offloading technique can be adopted at the cost of additional offloading overhead. Additionally, the choice between hierarchical and k-means clustering has negligible impact on performance.

**K-means MoE Routing.** In the ablation study, we design a variant that employs a learnable linear gating network for MoE routing. This variant exhibits suboptimal performance, possibly due to the

Table 5: Licenses for released assets

| Asset | License |
|---|---|
| FLUX.1 [dev] | Apache-2.0 license |
| Qwen2.5-VL | Apache-2.0 license |

under-utilization of certain experts. We further analyze the expert utilization pattern in this section and observe that several experts are underused, which is equivalent to using fewer experts than our full model. We visualize the expert assignment patterns of the MoE layers in the 1st and 2nd blocks of Mod-Adapter during training in Fig.6 (a) and Fig.6 (b), respectively. The training set contains 106,104 samples, and for each sample, Mod-Adapter predicts $N = 57$ modulation directions, resulting in 6,047,928 total inputs (106,104 × 57) to each MoE layer per training epoch. Each bar in Fig.6 indicates the percentage of inputs routed to each expert. As shown, Expert 7 (3.8%) and Expert 8 (4.1%) in the first Mod-Adapter block, as well as Expert 4 (1.1%) in the second block, receive significantly fewer inputs during training. This may lead to slower training of these experts, which in turn makes them less likely to be selected in subsequent iterations, resulting in their insufficient utilization. In contrast, our parameter-free $k$-means-based routing mechanism pre-defines expert assignments before training, based on k-means clustering over the neutral features of all concept words in the training dataset, helping ensure sufficient utilization of all experts throughout the training process as shown in Fig.6 (c).

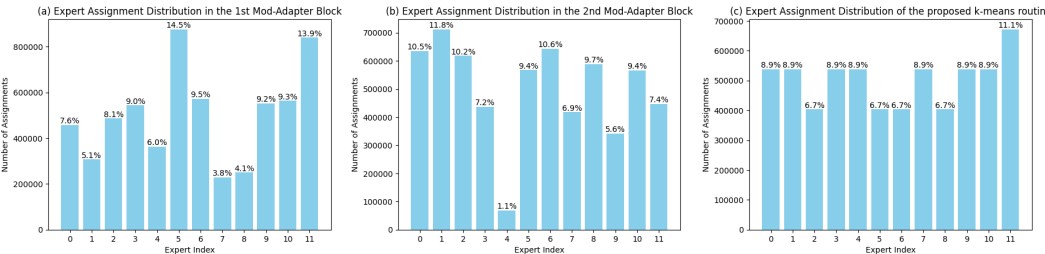

Figure 6: **Distribution of Expert Assignment.** Each bar indicates the percentage of inputs routed to each expert.

## G  ADDITIONAL QUALITATIVE RESULTS

In this section, we present additional qualitative results of our method (see Fig. 7, 8, 9, 11 and Fig. 12.)

## H  LICENSES FOR RELEASED ASSETS

This work uses the listed projects in Tab. 5 released under their licenses. We strictly adhered to their license requirements; the original projects' copyright notices and license texts can be found in their official repositories.

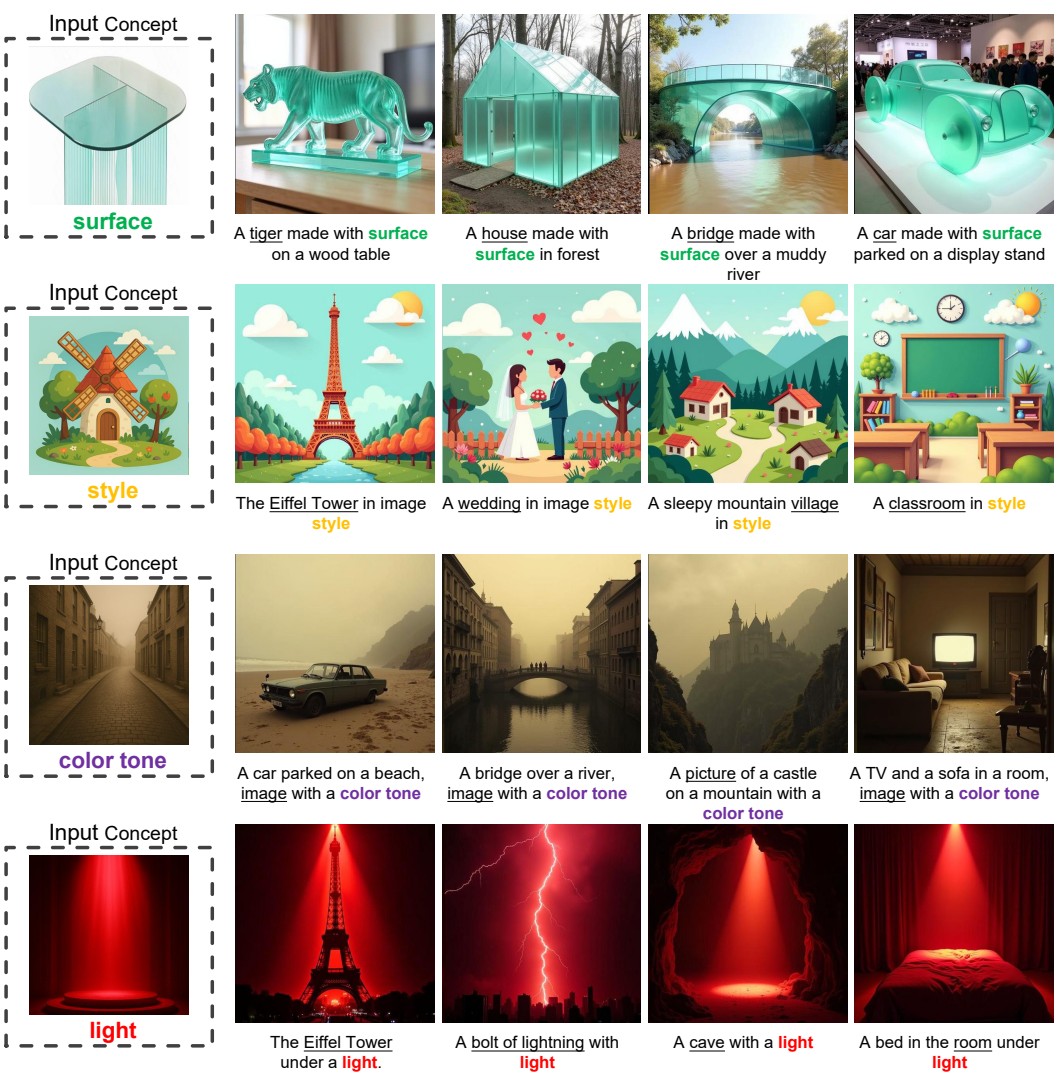

Figure 7: **More single-concept personalization results of our method.** Colored words in the prompt indicate concepts to be personalized; underlined text highlights elements reflecting prompt alignment.

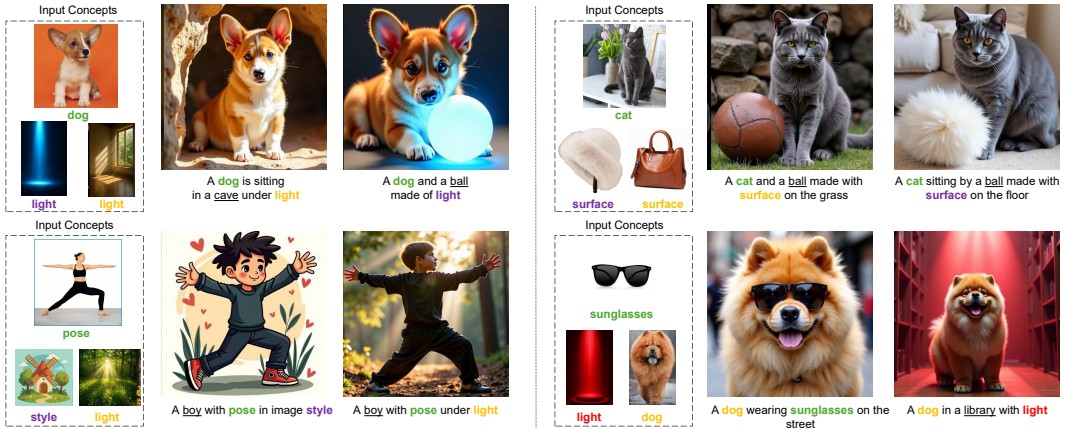

Figure 8: **More multi-concept personalization results of our method.** Colored words in the prompt indicate concepts to be personalized; underlined text highlights elements reflecting prompt alignment.

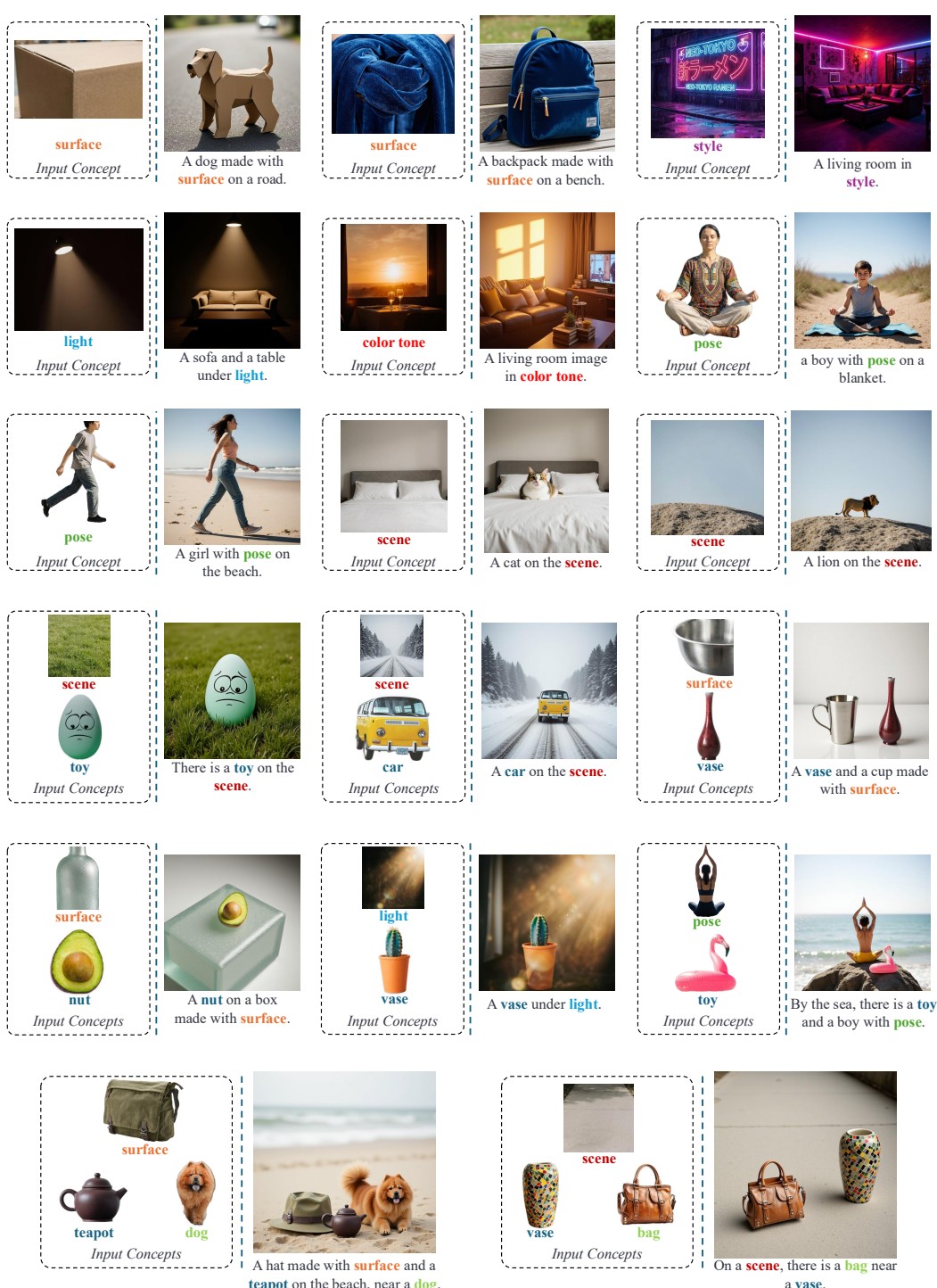

Figure 9: **More qualitative examples of our method on abstract concept personalization.** Colored words in the prompt indicate concepts to be personalized, including abstract concepts and their combinations with object concepts.

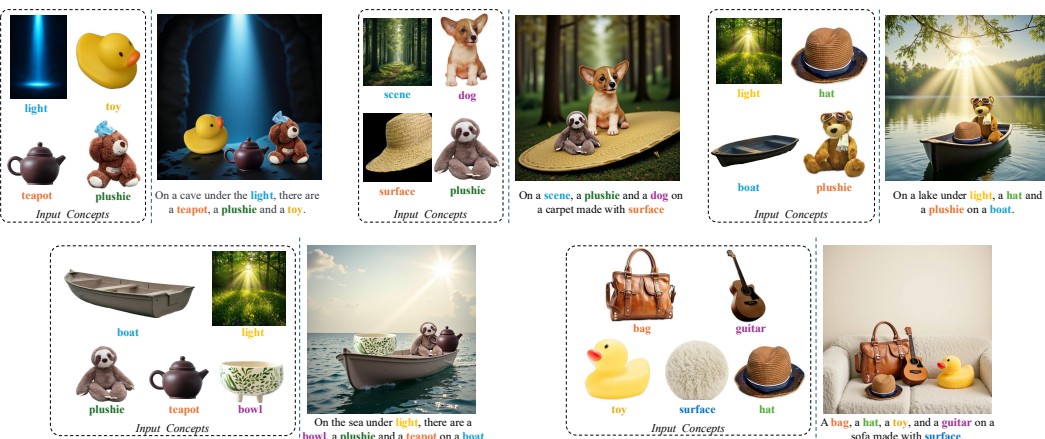

Figure 10: **The qualitative results of our method for customizing an extreme number of concepts (4 to 5 concepts).** Colored words in the prompt indicate concepts to be personalized.

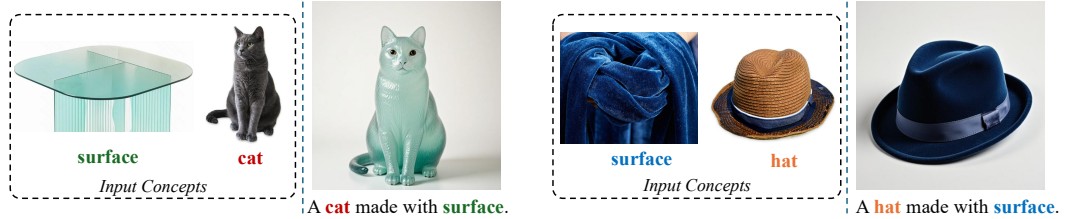

Figure 11: **Qualitative results of combining abstract concepts and concrete objects.** Our method effectively decouples abstract concepts and concrete objects and successfully recombines them. Colored words in the prompt indicate concepts to be personalized.

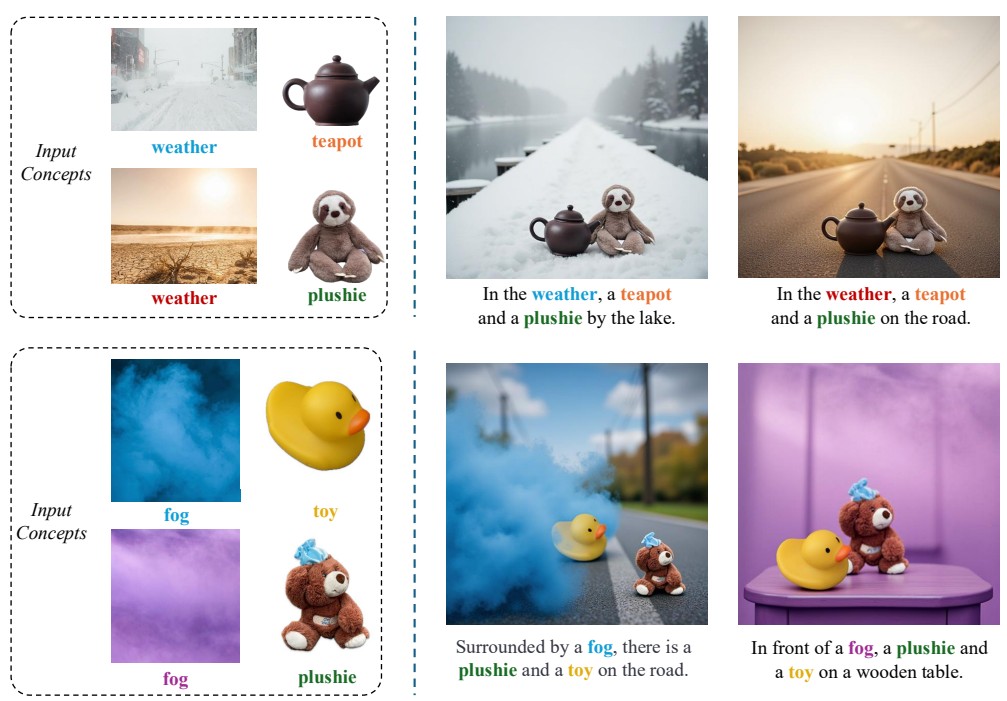

Figure 12: **Qualitative results demonstrating the generalization of our method to concept categories completely unseen during training.** Colored words in the prompt indicate concepts to be personalized.

