# OpenReview forum: "Mod-Adapter: Tuning-Free and Versatile Multi-concept Personalization via Modulation Adapter"
_ICLR.cc/2026/Conference — ICLR 2026 Poster_

### Official Review · Reviewer_Vhnb · 2025-10-28

**Soundness:** 3
**Presentation:** 3
**Contribution:** 2
**Rating:** 4
**Confidence:** 3

**Summary:**

The author propose a tunning-free personalization method for abstract concepts such as pose, lighting and surface by an innovative module Mod-Adapter. Also, they extend the DreamBench to form a new benchmark DreamBench-Abs for evaluating baselines and Mod-Adapter

**Strengths:**

- This work is the first to introduce the 'mod' concept from TokenVerse into tuning-free abstract concept personalization.
- This work uses a reasonable design for the Mod-Adapter and VLM-guided pre-training.
- The Mod-Adapter shows good results on sample examples.

**Weaknesses:**

- Limitation in qualitative results: This paper only shows a very limited number of abstract concept customization. Many abstract concepts are repeatedly displayed in different figures, which greatly reduces the reliability of the method's effectiveness. I am not sure the success rate of this method is beyond these examples. Can authors show more examples, even report the fail rate? If not, is it beacuse the distribution of training dataset?
- Insufficient motivation for using MoE: Authors discuss the reason for using MoE instead of simply MLP, do authors have any experiements to show how the mapping different between objects? Do the difference happen between objects (cat, dog, $\cdots$) and abstract concepts, or there are more complex principle for the mapping pattern gap? More discussion and experiments here might enhance the motivation for using MoE.

**Questions:**

- Is it possible to combine abstract concepts and concrete objects? The authors claim that this approach effectively decouples abstract concepts and concrete objects. So, for example, is it possible to achieve an effect like \<img1\> surface \<img2\> cat?

- What is the approximate proportion of abstract concept data in the training dataset? How many types of abstract concepts are inclued in the dataset?

Minor questions:

- Which VLM do authors use for evaluating CP and PF?

---

> ### Author Response · Authors · 2025-12-04
> **Point-by-Point Responses to Reviewer Vhnb**
>
> We thank the reviewer for all valuable comments. We provide point-by-point responses below.
> Additional qualitative results have been included in Appendix H (Section: QUALITATIVE RESULTS FOR THE REBUTTAL) of the revised submission.
>
> 1.**Show more qualitative results (W1)**. We have supplemented Figure 9 in Appendix H with more qualitative examples of abstract concept personalization. All these test examples were unseen in the training data, and the average failure rate is approximately 36%. Furthermore, we demonstrate the effectiveness of our method on personalization with an extreme number of concepts in Figure 10, and its generalization capabilities on completely unseen concept categories in Figure 12. These results further validate the reliability and effectiveness of our method.
>
>
>
> 2.**Clarification on the Motivation for Using MoE (W2).**
> (1) Adopting proof by contradiction to prove the intuition behind the MoE design that different types of concepts exhibit distinct mapping patterns. If all concept types shared the same mapping pattern, this mapping could be represented by a single MLP. However, as shown by the quantitative and qualitative results of the "w/o MoE" ablation variant (which uses a single MLP), the performance is significantly inferior. This failure negates the assumption of uniformity, thereby proving that distinct mapping patterns indeed exist between different concepts.
> (2) To further demonstrate how the mapping differs between concepts: We selected representative, high-performing concepts for each expert in the MoE, implying that the expert's modeling closely matches the true mapping patterns of these concepts. To show the differences, we flattened the weight matrices of the linear layers from these experts into vectors and calculated the pairwise cosine similarity between them. The resulting similarity table is shown below:
>
> | Similarity Table | bowl  | a can | backpack | hat   | clock | boat  | dog   | doll  | style    | light    | pose     | color    |
> | ---------------- | ----- | ----- | -------- | ----- | ----- | ----- | ----- | ----- | -------- | -------- | -------- | -------- |
> | **bowl**             | 1     | -0.01 | 0.01     | 0     | 0.01  | 0     | 0     | 0.01  | 0        | 0        | 0        | -0.01    |
> | **a can**            | -0.01 | 1     | 0        | -0.01 | 0     | 0     | 0.01  | 0     | -0.02    | 0        | 0.01     | 0        |
> | **backpack**         | 0.01  | 0     | 1        | 0     | -0.01 | 0     | 0     | -0.02 | 0        | 0.01     | 0        | 0        |
> | **hat**              | 0     | -0.01 | 0        | 1     | 0     | 0     | 0     | -0.01 | -0.01    | 0        | 0        | 0.01     |
> | **clock**            | 0.01  | 0     | -0.01    | 0     | 1     | 0     | -0.01 | 0     | 0        | 0        | 0        | -0.01    |
> | **boat**             | 0     | 0     | 0        | 0     | 0     | 1     | 0     | -0.01 | 0        | 0.01     | 0        | 0        |
> | **dog**              | 0     | 0.01  | 0        | 0     | -0.01 | 0     | 1     | 0     | 0        | -0.01    | 0        | 0        |
> | **doll**             | 0.01  | 0     | -0.02    | -0.01 | 0     | -0.01 | 0     | 1     | 0        | -0.01    | 0.01     | -0.01    |
> | **style**            | 0     | -0.02 | 0        | -0.01 | 0     | 0     | 0     | 0     | 1        | -0.01    | **0.01** | 0        |
> | **light**            | 0     | 0     | 0.01     | 0     | 0     | 0.01  | -0.01 | -0.01 | -0.01    | 1        | 0        | **0.01** |
> | **pose**             | 0     | 0.01  | 0        | 0     | 0     | 0     | 0     | 0.01  | **0.01** | 0        | 1        | 0        |
> | **color**            | -0.01 | 0     | 0        | 0.01  | -0.01 | 0     | 0     | -0.01 | 0        | **0.01** | 0        | 1        |
>
> As observed, the cosine similarities between the linear mapping vectors of these concepts are extremely low, with a maximum similarity of only 0.0124. This indicates that the mapping patterns of these concepts are distinct. Besides, we observe relatively higher mapping pattern similarity between certain abstract concepts (e.g., light and color) compared to others.
>
>
> 3.**Combining Abstract Concepts and Concrete Objects (Q1)**. Our method is fully capable of achieving this effect. As demonstrated in Figure 11 of Appendix H, we showcase the generated results for combinations such as < img1 > surface + < img2 > cat and < img1 > surface + < img2 > hat. These results demonstrate that our method effectively decouples abstract concepts and concrete objects, and successfully recombines them.
>
>
>
> 4.**Statistics of Abstract Concept Data (Q2)**. The approximate proportion of abstract concept data in our training dataset is 25.17%. Regarding the diversity of abstract concepts, the dataset includes 24 subdivided categories spanning six major classes.
>
>
> 5.**VLM for Evaluation (Q3)**. Following the experimental settings of most prior works, we adopt the GPT-4o multimodal LLM to calculate the CP and PF scores.

---

### Official Review · Reviewer_4ZPx · 2025-10-29

**Soundness:** 3
**Presentation:** 3
**Contribution:** 2
**Rating:** 6
**Confidence:** 3

**Summary:**

This paper addresses the limitations of existing multi-concept personalized text-to-image generation methods. The authors propose MOD-ADAPTER, a tuning-free framework that can effectively customize both object and abstract concepts, with the use of vision-language cross-attention, mixture-of-experts, and a VLM-guided pre-training strategy. The authors also extend the DreamBench benchmark to include abstract concepts and validate the method through quantitative and qualitative comparisons and user studies, showing state-of-the-art performance.

**Strengths:**

1.	The paper addresses the limitations of existing methods by proposing a tuning-free framework that can handle both object and abstract concepts. The extended benchmark is valuable for further studies.

2.	The integration of Vision-Language Cross-Attention and MoE layers in the MOD-Adapter module is well-justified and effective, proven in ablation studies.

3.	Experimental results underscore the effectiveness of the proposed method.

**Weaknesses:**

1.	While MOD-ADAPTER introduces a tuning-free improvement, the core idea of leveraging the DiT modulation space for multi-concept personalization builds on TokenVerse.

2.	The paper seems to claim its ability to deal with unseen concepts (neither used in training nor pre-training), however, the experiments are not split based on whether the concept is used in training. Besides, I doubt that some designs of the method, like the number of experts in MOE, may be highly dependent on the pre-training and training data.

3.	The main difference between cross-attention and the proposed Vision Language Cross-Attention is the usage of CLIP image encoder and text encoder, but in Figure 2, these two encoders are outside of the VL attention block, which is confusing.

4.	A common advantage of tuning-free frameworks is fast inference speed, which is also claimed in your paper. However, I don’t see any time complexity analysis in the experiments to prove it.

5.	There are misunderstanding typos in the paper, such as “w/o k-means” in Figure 4.

**Questions:**

1.	What’s the ratio of concepts in evaluation that are not used in pre-training and training? For these unseen concepts, please provide comparisons.

2.	Please refer to the weaknesses.

---

> ### Author Response · Authors · 2025-12-04
> **Point-by-Point Responses to Reviewer4ZPx**
>
> We thank the reviewer for all valuable comments. We provide point-by-point responses below.
>
> 1.**Discussion with TokenVerse (W1):**
> While both methods utilize the DiT modulation space, our work introduces a distinct tuning-free framework composed of a novel Mod-Adapter module and a VLM-guided pre-training strategy. The core difference lies in an innovative module, Mod-Adapter, which leverages the image-text alignment capability of CLIP to extract concept features and the adaptive capacity of MoE layers to map them into the modulation space. Besides, we introduce a novel pre-training strategy that leverages the strong image understanding capabilities of a frozen VLM to facilitate the alignment of concept image features with the DiT modulation space.
>
>
> 2.**Clarification on evaluation and training data (W2 & Q1)**.
> (1) 100% of the evaluation concepts were unseen in pre-training and training. All quantitative and qualitative experiments in the main paper were conducted on benchmarks completely outside the training distribution. As stated in Line 339, none of the test images or prompts overlap with the training data, which is consistent with the standard settings of previous tuning-free methods.
> (2)The number of experts in MoE is primarily determined by the 80GB GPU memory constraint, independent of the training data. As has been shown in the ablation study, increasing the number of experts increases model parameters and capacity, leading to better performance. 12 experts is the maximum number that fits within 80GB GPU memory. It can be further increased if using CPU offloading techniques, at the cost of inference time.
>
>
>
>
> 3.**Confusion regarding Figure 2 and typos in Figure 4 (W3 & W5).** Thanks for the suggestion. We have revised the Figure 2 and Figure 4 in the updated submission to avoid confusion.
>
>
> 4.**Time Complexity Analysis (W4)**. We report the inference time of different methods in the table below, measured on NVIDIA A800-SXM4 GPUs:
>
> | Method            | Inference Time (Single-Concept) | Inference Time (Multi-Concept) |
> |-------------------|---------------------------------|--------------------------------|
> | Emu2              | 11.55s                          | 11.73s                         |
> | MIP-Adapter       | 5.98s                           | 6.61s                          |
> | MS-Diffusion      | 4.63s                           | 5.01s                          |
> | TokenVerse        | 9.80s (+40min tuning)           | 28.93s (+40min tuning)         |
> | Mod-Adapter(Ours) | 4.95s                           | 6.98s                          |
>
> As shown, our method demonstrates inference speeds comparable to or faster than other tuning-free methods, while achieving significantly superior results in quantitative and qualitative comparisons. In contrast, the tuning-based method TokenVerse requires fine-tuning a separate model for each distinct concept at test time, which incurs an additional overhead of approximately 40 minutes.

---

### Official Review · Reviewer_5SAg · 2025-11-01

**Soundness:** 3
**Presentation:** 4
**Contribution:** 3
**Rating:** 6
**Confidence:** 4

**Summary:**

This paper presents Mod-Adapter, a tuning-free framework for versatile multi-concept personalization in text-to-image generation that can customize both object and abstract concepts without test-time fine-tuning. The method leverages the localized and semantically meaningful properties of the modulation space in pre-trained Diffusion Transformers (DiTs). The key contributions include: (1) a novel Mod-Adapter module that predicts concept-specific modulation directions using vision-language cross-attention for feature extraction and Mixture-of-Experts (MoE) layers for mapping to modulation space; (2) a VLM-guided pre-training strategy to facilitate training; and (3) an extended benchmark DreamBench-Abs incorporating abstract concepts. Experimental results show state-of-the-art performance in multi-concept personalization.

**Strengths:**

- The paper addresses an important and underexplored challenge in multi-concept personalization - the simultaneous customization of both object and abstract concepts without test-time fine-tuning. This represents a significant practical limitation of existing methods that the authors successfully tackle.
- The insight to leverage the localized and semantically meaningful properties of the DiT modulation space is particularly clever. This approach enables the additive combination of multiple concept modulations without interference, which is crucial for multi-concept personalization.
- The creation of DreamBench-Abs by incorporating abstract concepts into the standard benchmark is a valuable contribution to the research community. This enables more rigorous evaluation of methods claiming to handle diverse concept types.
- The authors provide extensive quantitative, qualitative, and human evaluation results demonstrating the superiority of their method over state-of-the-art approaches. The ablation studies effectively validate the contribution of each proposed component.
- The tuning-free nature of the approach makes it highly practical for real-world applications where test-time fine-tuning is infeasible or impractical. The ability to handle both object and abstract concepts significantly expands the range of possible applications.

**Weaknesses:**

- The Mod-Adapter contains 1.67B parameters, which is large for an adapter module. This raises questions about the practical deployment of the method, especially in resource-constrained environments. The paper does not adequately address this concern.
- The paper does not sufficiently explore the generalization capabilities of the method to unseen concept types or combinations. The ablation studies focus on component removal but lack analysis of performance under varying conditions.
- The paper mentions that performance degrades with more than 3 concepts, but does not provide a detailed analysis of this limitation or potential solutions. This represents a significant practical constraint.

**Questions:**

1. What is the computational overhead of the method during inference compared to baseline approaches? Have the authors measured inference time and memory usage?
2. The paper mentions performance degradation with more than 3 concepts. Could the authors analyze this limitation more thoroughly? Are there specific types of concepts that are more prone to causing interference?
3. How does the method handle concept drift or domain shift? For example, would it maintain performance when applied to concepts from domains not represented in the training data?
4. Could the authors provide more insights into the MoE routing mechanism? Why does k-means clustering perform better than learnable gating networks? How is balanced expert utilization ensured?
5. What is the effect of varying the number of DiT blocks (N) on performance? The paper uses all 57 blocks from FLUX, but would a subset be sufficient?

---

> ### Author Response · Authors · 2025-12-04
> **Point-by-Point Responses to Reviewer 5SAg (1/2)**
>
> We thank the reviewer for all valuable comments. We provide point-by-point responses below.
> Additional qualitative results have been included in Appendix H (Section: QUALITATIVE RESULTS FOR THE REBUTTAL) of the revised submission.
>
> 1.**Addressing the concern of practical deployment in resource-constrained environments (W1)**:
> We implemented FP8 quantization on the Mod-Adapter weights.
> This optimization successfully reduced the GPU memory usage during multi-concept personalization inference from 47.5GB to 38.1GB, with only negligible performance degradation, as shown in the table below.
>
> | Multi-Concept | CP↑  | PF↑  | CP·PF↑ | CLIP-T↑ |
> | ------------- | ---- | ---- | ------ | ------- |
> | Ours          | 0.70 | 0.89 | 0.62   | 0.330   |
> | Ours (FP8)    | 0.67 | 0.88 | 0.59   | 0.328   |
>
>
> To further reduce memory for deployment in even more resource-constrained environments, future implementations can adopt an FP8-quantized FLUX DiT or other efficient architectures like SANA DiT [1] as the backbone.
>
> [1] SANA: Efficient High-Resolution Image Synthesis with Linear Diffusion Transformers. ICLR 2025.
>
>
>
> 2.**Generalization capabilities to unseen concept types or combinations (W2.1 & Q3)**.
> (1) Generalization to Unseen Combinations: Our multi-concept personalization evaluation inherently tests generalization to unseen combinations. As stated in Line 678 of the main paper, the model used for all quantitative and qualitative experiments was trained exclusively on single-concept data. Therefore, its strong performance on multi-concept personalization demonstrates its ability to robustly generalize to concept combinations never seen during training. (2) Generalization to Unseen Types: Our method also demonstrates generalization capabilities on completely unseen concept categories. While existing evaluation protocols typically focus on known categories, we conducted additional experiments on concept categories unseen during training (e.g., fog and weather). We achieved comparable quantitative results as shown in the table below and present corresponding qualitative results in Figure 12 of Appendix H.
>
>
> | Multi-Concept   | CP↑  | PF↑  | CP·PF↑ | CLIP-T↑ |
> | --------------- | ---- | ---- | ------ | ------- |
> | Ours (Standard) | 0.70 | 0.89 | 0.62   | 0.330   |
> | Ours (Unseen)   | 0.68 | 0.88 | 0.60   | 0.329   |
>
> Compared to the evaluation following the standard setting (denoted as "Ours (Standard)"), our method maintains robust generalization performance on unseen concept categories denoted as "Ours (Unseen)". This capability stems from the proposed Mod-Adapter leveraging the generalizable vision-language alignment capabilities of the CLIP model.
>
>
> 3.**More Ablation Performance Analysis (W2.2)**: We have refined Section 4.3 (Ablation Study) in the updated submission. Specifically, we have expanded this section with detailed explanations and analyses regarding the performance variations under varying conditions.

---

> ### Author Response · Authors · 2025-12-04
> **Point-by-Point Responses to Reviewer 5SAg (2/2)**
>
> 4.**Addressing the limitation when customizing more than three concepts (W3 & Q2)**:
> - Analysis:
> At the time of submission, our model was trained only on single-concept data, which is the primary cause for the performance drop in multi-concept customization with an extreme number of concepts. Additionally, our framework relied solely on the CLIP image encoder for image feature extraction, which captures high-level features but tends to lose low-level details of object-type concepts.
>
> - Potential Solutions:
> Due to time constraints in the rebuttal stage, we synthesized training data involving two concepts (abstract and object) using the FLUX model, and combined it with the open-source MuSAR-Gen [1] multi-concept dataset, resulting in a total of 52,000 multi-concept samples for training.
> Furthermore, to mitigate the loss of low-level object details, we concatenated the VAE latents of concept images with the noise latents as additional conditions.
> We also introduced UnoPE [2] positional encoding to distinguish VAE latents from multiple reference images and applied lightweight LoRA fine-tuning to the joint attention layers in DiT.
>
> - Experimental Results:
> We present qualitative results for customizing 4 to 5 concepts in Figure 10 of Appendix H, and quantitative results in the table below.
>
>
> | Multi-Concept       | CP↑  | PF↑  | CP·PF↑ | CLIP-T↑ |
> | ------------------- | ---- | ---- | ------ | ------- |
> | Ours (2~3 concepts) | 0.70 | 0.89 | 0.62   | 0.330   |
> | Ours (4~5 concepts) | 0.65 | 0.87 | 0.57   | 0.328   |
>
> As shown, after training on multi-concept data and incorporating VAE latents of concept images, our method achieves performance on extreme combinations (4-5 concepts) comparable to scenarios with fewer concepts ($\leq 3$). Future work can focus on curating training data with even more concepts to further extend the maximum number of concepts supported by the model.
>
> [1]. MUSAR: Exploring Multi-Subject Customization from Single-Subject Dataset via Attention Routing. ArXiv:2505.02823.
>
> [2]. Less-to-More Generalization:  Unlocking More Controllability by In-Context Generation. ICCV 2025.
>
>
> 5.**Comparison of Computational Overhead during Inference (Q1)**.
> We report the inference time and GPU memory usage of different methods, measured on NVIDIA A800-SXM4 GPUs, in the table below.
>
> | Method            | Time (Single-Concept) | Memory (Single-Concept) | Time (Multi-Concept) | Memory (Multi-Concept) |
> |-------------------|-----------------------|-------------------------|----------------------|------------------------|
> | Emu2              | 11.55s                | 80GB                    | 11.73s               | 80GB                   |
> | MIP-Adapter       | 5.98s                 | 19GB                    | 6.61s                | 19GB                   |
> | MS-Diffusion      | 4.63s                 | 14GB                    | 5.01s                | 14GB                   |
> | TokenVerse        | 9.80s                 | 57GB                    | 28.93s               | 57GB                   |
> | Mod-Adapter(Ours) | 4.95s                 | 45GB                    | 6.98s                | 48GB                   |
>
>
> As shown in the table, our method demonstrates comparable or even faster inference speeds compared to baseline approaches. Despite this similar computational cost, we achieve significantly superior results in both quantitative and qualitative evaluations. Furthermore, our method is efficient in terms of GPU memory consumption and can be further optimized for deployment in resource-constrained environments (as detailed in our response to W1 above).
>
>
> 6.**Further Explanations on the MoE Routing Mechanism (Q4)**:
> (1) As analyzed in Appendix F (MORE ABLATION ANALYSIS), employing a learnable linear gating network for MoE routing tends to result in the under-utilization of certain experts. This phenomenon reduces the model's capacity, making it equivalent to using fewer experts than designed, which leads to suboptimal performance. In contrast, our parameter-free k-means-based routing mechanism pre-defines expert assignments before training based on k-means clustering over the neutral features of all concept words, helping ensure sufficient utilization of all experts throughout the training process.
> (2)
> To ensure balanced expert utilization, we adopt Balanced K-Means clustering — which augments the standard K-Means objective with an additional cluster-size constraint to enforce an approximately equal number of samples per cluster. This corresponds to assigning a similar number of concept categories to each expert.
>
> 7.**Effect of Using a Subset of DiT Blocks (Q5)**. Thanks for the suggestion. We experimented with applying the proposed Mod-Adapter only to the 19 double-stream DiT blocks of FLUX (instead of all 57) and observed negligible performance degradation. This indicates that applying the proposed modulation only in the double-stream blocks is sufficient, inspiring more efficient designs in future work.

---

### Official Review · Reviewer_Bc9H · 2025-11-01

**Soundness:** 3
**Presentation:** 3
**Contribution:** 2
**Rating:** 4
**Confidence:** 4

**Summary:**

This paper proposes a tuning-free multi-concept personalized image generation method called Mod-Adapter, which can simultaneously customize specific objects and abstract visual concepts. Based on the modulation space of the pre-trained Diffusion Transformer, the method predicts concept-specific modulation directions via lightweight adapters and incorporates vision-language cross-attention and MoE mechanisms for efficient feature mapping. The authors also design a VLM-guided pre-training strategy to mitigate training difficulties and extend the evaluation benchmark DreamBench-Abs.

**Strengths:**

- Achieving tuning-free multi-concept personalization. Unlike previous methods, such as TokenVerse that require fine-tuning small MLPs for each new concept, Mod-Adapter is trained once to be universally applicable to all concepts. During inference, only reference images and concept words need to be input without any optimization steps.
- Extensive experiments are conducted on several datasets. The proposed method achieves good performance in multi-concept personalization.

**Weaknesses:**

- The proposed Mod-adaptor is depedent on the training data. Mod-Adapter needs to be trained on datasets containing abstract concepts, synthetic data + MVImgNet + AFHQ. If the user concepts fall outside the training distribution, its generalization ability is questionable.
- The performance degrades when customizing more than three concepts simultaneously, which limits its application in extremely complex scenarios.
- Why in Table 1, the performance of Mod-adaptor is 0.61, which is lower than previous Emu2 method?

**Questions:**

See above.

---

> ### Author Response · Authors · 2025-12-03
> **Point-by-Point Responses to Reviewer Bc9H**
>
> We thank the reviewer for all valuable comments. We provide point-by-point responses below.
> Additional qualitative results have been included in Appendix H (Section: QUALITATIVE RESULTS FOR THE REBUTTAL) of the revised submission.
>
> 1.**The generalization ability on concepts outside the training distribution (W1):**
> (1) Generalization to Unseen Instances: All quantitative and qualitative experiments in the main paper were already conducted on benchmarks outside the training distribution. This includes the standard DreamBench and our extended abstract concepts. As stated in Line. 339, none of the test images or prompts overlap with our training data, which is consistent with the experimental settings of previous tuning-free methods.
> (2) Generalization to Unseen Categories: Our method also demonstrates generalization capabilities on completely unseen concept categories. While existing evaluation protocols typically focus on known categories, we conducted additional experiments on concept categories unseen during training (e.g., fog and weather). We achieved comparable quantitative results as shown in the table below and present corresponding qualitative results in Figure 12 of Appendix H.
>
>
> | Multi-Concept   | CP↑  | PF↑  | CP·PF↑ | CLIP-T↑ |
> | --------------- | ---- | ---- | ------ | ------- |
> | Ours (Standard) | 0.70 | 0.89 | 0.62   | 0.330   |
> | Ours (Unseen)   | 0.68 | 0.88 | 0.60   | 0.329   |
>
> Compared to the evaluation following the standard setting (denoted as "Ours (Standard)"), our method maintains robust generalization performance on unseen concept categories denoted as "Ours (Unseen)". This capability stems from the proposed Mod-Adapter leveraging the generalizable vision-language alignment capabilities of the CLIP model.
>
>
> 2.**Addressing the limitation when customizing more than three concepts (W2)**:
> - Analysis:
> At the time of submission, our model was trained only on single-concept data, which is the primary cause for the performance drop in multi-concept customization with an extreme number of concepts. Additionally, our framework relied solely on the CLIP image encoder for image feature extraction, which captures high-level features but tends to lose low-level object details.
>
> - Potential Solutions:
> Due to time constraints in the rebuttal stage, we synthesized training data involving two concepts (abstract and object) using the FLUX model, and combined it with the open-source MuSAR-Gen [1] multi-concept dataset, resulting in a total of 52,000 multi-concept samples for training.
> Furthermore, to mitigate the loss of low-level object details, we concatenated the VAE latents of concept images with the noise latents as additional conditions.
> We also introduced UnoPE [2] positional encoding to distinguish VAE latents from multiple reference images and applied lightweight LoRA fine-tuning to the joint attention layers in DiT.
>
> - Experimental Results:
> We present qualitative results for customizing 4 to 5 concepts in Figure 10 of Appendix H, and quantitative results in the table below.
>
>
> | Multi-Concept       | CP↑  | PF↑  | CP·PF↑ | CLIP-T↑ |
> | ------------------- | ---- | ---- | ------ | ------- |
> | Ours (2~3 concepts) | 0.70 | 0.89 | 0.62   | 0.330   |
> | Ours (4~5 concepts) | 0.65 | 0.87 | 0.57   | 0.328   |
>
> As shown, after training on multi-concept data and incorporating VAE latents of concept images, our method achieves performance on extreme combinations (4-5 concepts) comparable to scenarios with fewer concepts ($\leq 3$). Future work can focus on curating training data with even more concepts to further extend the maximum number of concepts supported by the model.
>
> [1]. MUSAR: Exploring Multi-Subject Customization from Single-Subject Dataset via Attention Routing. ArXiv:2505.02823.
>
> [2]. Less-to-More Generalization:  Unlocking More Controllability by In-Context Generation. ICCV 2025.
>
>
> 3.**Clarification on Single-Concept CP Scores in Table 1 (W3):**
> As analyzed in Lines 361-364 and 367-371 of main paper,
> Emu2 fails to disentangle the abstract concept from the object.
> Consequently, it simply replicates the original object in the generated image, producing incorrect objects (e.g., generating a handbag instead of the requested wallet).
>  This observation aligns with its high Concept Preservation (CP) scores—which in this context reflect mere replication—but very low Prompt Fidelity (PF) scores. Therefore, the product of these two metrics (CP·PF) is recommended as a more comprehensive evaluation of the model's overall performance. As shown in Table 1, Emu2's CP·PF score is significantly lower than ours, demonstrating the superior balance of our method between concept preservation and text alignment.

---

### Meta-Review · Area_Chair_gRYU · 2026-01-06

**Summary:**

This paper received borderline reviews. The main concerns raised in the reviews are:
1. generalization beyond concept categories seen during training is unclear (`Bc9H`, `5SAg`).
2. absence of customization results with more than three concepts (`Bc9H`, `5SAg`).
3. limited examples of abstract concept customization (`Vhnb`).
4. no failure analysis (`Vhnb`).
5. missing analysis on computational cost (`5SAg`, `4ZPx`).
6. insufficient motivation/insights on the MoE mechanism (`5SAg`, `4ZPx`, `Vhnb`).

Overall, most of the concerns have been addressed. The remaining issues seem fixable in the final revision. While the contributions do not seem ground-breaking, the paper does a good job in demonstrating substantial improvement over existing baselines. Therefore, the AC recommends Accept.

**Reviewer Concerns:**

1. Concern #1 is partially addressed by the new examples with unseen concepts (`fog` and `weather`) shown in Fig. 12. A more thorough analysis of the generalization capabilities is still missing.
2. Concern #2 is mostly addressed by the new results with 4-5 concepts shown in Fig. 10, after training on multi-concept data. While the overall customization quality is decent, the realism drops slightly in some examples (eg, relative size, position, and illumination of the objects). Moreover, this is also constrained by the difficulty in curating large-scale multi-concept datasets.
3. Concern #3 is addressed by the new results in Fig. 9.
4. Concern #4 has not been fully addressed. The authors reported a single number of 36% failure rate in the rebuttal, but it is unclear how this number is obtained. A more informative failure analysis is still missing.
5. Concern #5 is addressed by the new results on computational analysis.
6. Concern #6 is addressed by the new ablation analysis and motivation elaboration.

**Reviewer Scores:**

1. Reviewer `Bc9H` (4->6): it is likely that the reviewer would increase the rating from 4 to 6 as their concerns have mostly been addressed.
2. Reviewer `5SAg` (6->6+): it is likely that the reviewer would maintain or increase their rating.
3. Reviewer `4ZPx` (6->6+): it is likely that the reviewer would maintain or increase their rating.
4. Reviewer `Vhnb` (4->?): it is difficult to predict whether the reviewer would be satisfied enough with the rebuttal to increase their rating, as their concern (#4) on missing failure analysis still remains.

---

### Decision · Program_Chairs · 2026-01-26

Accept (Poster)